# Characterization, Recombinant Production, and Bioactivity of a Novel Immunomodulatory Protein from *Hypsizygus marmoreus*

**DOI:** 10.3390/molecules28124796

**Published:** 2023-06-15

**Authors:** Shuhui Yu, Ying Wang, Yingying Wu, Dapeng Bao, Wei Bing, Yan Li, Hongyu Chen

**Affiliations:** 1School of Chemistry and Life Sciences, Changchun University of Technology, Changchun 130012, China; 2National Engineering Research Center of Edible Fungi, Ministry of Science and Technology (MOST), Key Laboratory of Edible Fungi Resources and Utilization (South), Ministry of Agriculture, Institution of Edible Fungi, Shanghai Academy of Agricultural Sciences, Shanghai 201403, China; 3Key Laboratory of Bionic Engineering, Ministry of Education, Jilin University, Changchun 130022, China

**Keywords:** *Hypsizygus marmoreus*, fungal immunomodulatory protein, protein structure, phylogenetic tree, recombinant expression, cytokine release

## Abstract

A novel fungal immunomodulatory protein (FIP), identified as FIP-hma, was discovered in the genome of an edible mushroom *Hypsizygus marmoreus*. Bioinformatics analysis suggested FIP-hma contained the cerato-platanin (CP) conserved domain and was categorized into Cerato-type FIP. In phylogenetic analysis, FIP-hma was clustered into a new branch of the FIP family, displaying large system divergence from most of the other FIPs. The higher gene expression of FIP-hma was observed during the vegetative growth stages than that during the reproductive growth stages. In addition, the cDNA sequence of FIP-hma was cloned and successfully expressed in *Escherichia coli* (*E. coli*) BL21(DE3). The recombinant protein of FIP-hma (rFIP-hma) was neatly purified and isolated by Ni-NTA and SUMO-Protease. The iNOS, IL-6, IL-1β, and TNF-α levels of RAW 264.7 macrophages were upregulated by rFIP-hma, indicating its activation of an immune response by regulating central cytokines. No cytotoxic effects were observed in an MTT test. The findings of this work discovered a novel immunoregulatory protein from *H. marmoreus,* provided a systematic bioinformatic profile, suggested an effective approach for its heterologous recombinant production, and reported its potent immunoregulatory activity in macrophages. This study sheds light on the physiological function research of FIPs and their further industrial utilization.

## 1. Introduction

*Hypsizygus marmoreus,* also known as shimeiji, bunashimeji, or beech mushroom, is a popular edible and medicinal mushroom in Asian vegetable markets [1]. They typically develop white or light brown umbrella-shaped caps in morphology. In previous studies, many bio-active ingredients were reported from mushrooms, such as polysaccharides, proteins, terpenoids, and low molecule weight compounds [2,3,4], which display potent physiological effects such as anticancer [5], antioxidative [6], anti-inflammatory [7], and asthma amelioration [8]. The polysaccharides extracted from the mycelia of *H. marmoreus* showed antioxidant and anti-pulmonary inflammation functions, whose mechanism of action might be attributed to the down-regulation of the serum complement 3 (C3), glutamyl transpeptidase (GGT), and high-sensitivity C-reactive protein (hs-CPR) [7]. A fucomannogalactan (FMG-Hm) obtained from fruiting bodies of *H. marmoreus* inhibited colony formation and cell migration of murine melanoma cells without cytotoxicity, its structure was found to consist of α-(1→6)-linked galactopyranosyl main chain [9]. Nevertheless, little is known about the bioactive proteins from *H. marmoreus* and other mushrooms.

Fungal immunomodulatory proteins (FIPs) are a new class of bioactive proteins reported in fungi, including those popular edible and medicinal mushrooms on the market, such as *G. lucidum* (LZ-8 or FIP-glu) [10], *F. velutipes* (FIP-fve) [11], *V. volvacea* (FIP-vvo) [12], and *G. sinensis* (FIP-gsi) [13]. The bioactive effects of FIPs have attracted research interest since their discovery, such as hemagglutination, anti-allergy, anti-inflammation, immunomodulation, and anti-tumor [14,15]. The immune modulators originating from mushrooms stimulate both the innate immune system and the acquired immune system [2]. FIP-fve inhibits systemic anaphylaxis reactions and was found to enhance the transcriptional expression of IL-2 and IFN-γ [11]. FIPs were proven to be T cell activators, stimulating many important cytokines (IL-2, IL-3, TNF-α, and IFN-γ, etc.) [16]. FIP-gsi upregulates the expression of IL-2, IL-3, IL-4, INF-γ, TNF-α, and IL-2 receptor (IL-2R) in mouse spleen cells [13]. 

Macrophages are multifunctional cells that play a principal role in defending microorganisms, clearing aging or dead cells, and repairing injured tissues [17]. Macrophages serve as initiators of adaptive immune response in various diseases [18]. Activation of other immune cells and the downstream cascade is one of the main functions of macrophages. One mechanism by which macrophages attract and activate other immune cells is the production and release of soluble factors such as IL-1 β, IL-6, IL-12, and TNF- α [19]. The low yield and high cost of extracting and isolating natural FIPs from edible fungi are the main factors limiting the application of FIPs. According to previous reports, LZ-8 was recombinantly expressed in the *Pichia pastoris* system and exhibited a similar immune regulation ability to natural proteins [20]. FIP-fve was successfully expressed in *E. coli*, whose yield reached 29.1 mg/L with 97.1% purity; rFIP-fve regulated IL-2 and IFN-γ in mice, showing potential for industrial applications [21]. A recombinant fungal immunomodulatory protein from *Lignosus rhinoceros* was produced in *P. pastoris* and *E. coli* host expression systems, which (at 4–8 μg/mL doses) induced expression of Th1 (IFN-γ, TNF-α) and Th2 (IL-6, IL-4, IL-5, IL-13) cytokines in mice splenocytes [22]. A recombinant fungal immunomodulatory protein, GMI, was cloned from *G. microsporum* and purified, which inhibited migration and invasion of lung cancer cells, the mechanism was related to restraining activation of the EGFR and Akt pathway [23]. 

In this study, gene mining was used to identify FIPs in *H. marmoreus*. The identified FIP was named FIP-hma and is reported for the first time. Further, a bioinformatics analysis was performed for FIP-hma to predict the structural and biochemical characteristics. Real-time PCR was used to determine the expression levels of FIP-hma during the growth and development of *H. marmoreus*. Further, the recombinant protein of FIP-hma (rFIP-hma) was successfully expressed in *E. coli* and isolated with the optimized conditions. The immune regulation activity of the purified rFIP-hma was tested in macrophages, including Western blots for cytokine expression and an ELISA assay for their release. Therefore, this study identified and characterized a novel immunoregulatory protein in *H. marmoreus*, providing a reference for further functional protein research and product development.

## 2. Results and Discussion

### 2.1. Structural and Biochemical Properties of FIP-hma

A novel FIP protein, FIP-hma, was identified in the *H. marmoreus* genome by local BLAST searches using an amino acid sequence of c13717 from *Ganoderma australe* as a bait. FIP-hma was a hydrophobic (Appendix A) small protein consisting of 145 amino acids (14.79 kDa). It was considered stable with an instability index of 22.85, referring to Hao et al.’s research that classifies proteins with an index > 40 as unstable [24]. The half-life of FIP-hma in mammalian reticulocyte, yeast, and *E. coli* was 30, >20, and >10 h, respectively. FIP-hma contained more hydrophobic amino acids as compared with hydrophilic amino acids. Ala, Leu, Gly, and Val were the most abundant amino acids in FIP-hma. In this work, we applied a classic FIP—*Ganoderma australe*—as the bait, which is a typical Cerato-type FIP with a signal peptide at the N-terminal and has high homology with the protein reported for *A. camphorate* (ACA) [25]. Cerato-type FIPs enhanced phagocytosis activity and CD86 (B7-2) expression as well as induced TNF-R and IL-1β production within murine peritoneal macrophages [26]. These results leave clues for us to explore the structure and function of the newly discovered FIP-hma.

As analyzed by Clustal X2 multiple sequence alignment and conserved structure, FIP-hma contained CP conserved domains and was clustered into Cerato-type FIPs, which is similar to the FIPs c13717, YZP, and ACA (Figure 1A) [14]. According to the molecular evolutionary relationship acquired by MEGAX phylogenetic tree construction, FIP-hma, c13717, YZP, and ACA formed a high level of support, with a unique, separate lineage, suggesting large system divergence between the FIP-hma and all other FIPs (Figure 1B). 

Generally, FIPs are classified into five typical subgroups based on their conserved structure and protein identity. Among them, the dominant category is the Fve-type FIPs containing the assigned Pfam Fve family domain signature PF09259 [14]. Nevertheless, the FIP-hma identified and modeled in this work was not a most-reported Fve-type FIP but rather a Cerato-type FIP marked by the representative domain Pfam PF07249 [14,26]. Evolutionarily, the FIP-hma was conversed from the other Cerato-type FIPs such as ACA [26] and YZP [27], and showed high homology. The findings of this study add information to understanding the structural basics of FIPs.

As predicted by AlphaFold, modeled FIP-hma formed spheroid-like structures containing short α-helixes, six β-sheets, and many random coils (Figure 2). The six β-sheets of FIP-hma might be more stable cores folded into two vertical layers. The three-dimensional structures of FIP-hma were different from those of Fve-type FIPs [14], the latter typically includes the N-terminal domain beginning with an N-terminal α-helix, and a sandwich-type fibronectin III (FNIII) C-terminal domain consisting of seven to nine β-sheets [28,29]. The FIP-hma protein had no typical α-helices at the N-terminus and fibronectin III domain at the C-terminus. Instead, it formed spheroid-like structures similar to ACA and YZP, which contained short α-helixes and three (ACA) or five β-sheets (YZP). The establishment of the three-dimensional structure model further demonstrated the inference based on the sequence domains and conservative motifs.

### 2.2. Real-Time RT-PCR Detection of FIP-hma Gene Expression Levels during the Development of H. marmoreus

As analyzed by real-time RT-PCR, using *FIP-hma* gene expression in the mycelium stimulation stage as a control, *FIP-hma* was expressed abundantly during the vegetative growth stages but was barely expressed during the reproductive growth stages, among which expression at completion of the post-ripening reached its peak (Figure 3A). In the reproductive growth stages, the expression of *FIP-hma* in substrate (containing mycelium) was higher than that in the fruiting body (Figure 3B). Taking together, the determined expression level of *FIP-hma* was higher during the vegetative growth stages (in the format of mycelia) than the reproductive growth stages (in the format of fruit bodies). In fungi, there are bioactive proteins reported to be largely expressed in the mycelial stage yet significantly reduced in the reproductive growth stage. For instance, laccases regulating fungal growth and mycelial metabolism were found to increase during the mycelial stages and began to decline during later stages in many edible fungi such as *V. volvacea* [30] and *H. marmoreus* [31]. Similarly, *FIP-hma* was inferred to participate in regulating the synthesis, transportation, and metabolite accumulation as well as host defense during mycelial growth, hence it might be observed with an analogous expression pattern as reported for laccase. The data of this work shows the expression of *FIP-hma* was more abundant in the mycelia of *H. marmoreus*, which provides a hint that liquid fermentation might be a more effective strategy for their protein production. Liquid fermentation has shown many advantages. It has a more efficient space utilization rate, saves land, has higher production capacity, and allows for annual stable production [32]. 

### 2.3. Expression of rFIP-hma in E. coli

The *E. coli* system was adopted to test the heterologous expression of rFIP-hma and hence to provide a potential approach for more effective protein production. pSmartI-FIP-hma was successfully expressed in large quantities in *E. coli* (Figure 4A). 

The addition of the IPTG (Isopropyl-β-d-thiogalactoside) inducer remarkably increased the expression of pSmartI-FIP-hma (Figure 4B). As compared with the control without IPTG induction, the expression of the target protein was largely elevated when treated with 0.2 mM or 1.0 mM IPTG at 15 °C or 37 °C (Lanes 1–4, Figure 4B). The sequence design for rFIP-hma includes the SUMO tag to promote increased protein solubility. SUMO (small ubiquitin-related modifier) is a popular solubility label in recent years that provides increased levels of proteins with soluble expression in *E. coli* and allows rapid purification of proteins of interest [33]. Kuo et al.’s research suggests that SUMO may have a chaperone-like function to help its fusion partners fold appropriately and maintain solubility [34]. To distinguish the protein included in the inclusion body of the host cell and the soluble proteins that are available for further separation and utilization, the soluble fraction of the expression product was separated, then sediment samples and supernatant samples were analyzed separately. When treated with 0.2 mM or 1.0 mM IPTG under 37 °C, the insoluble proteins existing in the inclusion body displayed strong bands but the expression of soluble proteins was very weak (Lanes 6–9, Figure 4B). Under the condition of 15 °C and 1.0 mM IPTG induction, though many proteins were left in the sediment, the target protein was more distributed in the supernatant, indicating more soluble proteins (Lanes 10–11, Figure 4B). Under the treatment of 0.2 mM IPTG at 15 °C, induced expression was inconspicuous (Lanes 12–13, Figure 4B). Therefore, to enhance the yield of soluble protein for further bioactivity research, the addition of 1.0 mM IPTG at 15 °C was selected as the expression condition.

At present, genome mining has become a new tool for searching for potential FIPs. Ten FIPs, such as FIP-lrh [22], FIP-mco [35], and FIP-sch2 [36], have been directly cloned through genome mining. The low yield and high cost limit the extraction and application of natural FIPs. The production of bioactive proteins in sufficient amounts is essential for their subsequent research or use as pharmaceutical ingredients. In this work, *E. coli* is used as a tool for the heterologous expression of FIP-hma. The *E. coli* system is considered the host of choice for recombinant protein production due to fast growth speed, easy operation, and low cost, whose associated parameters (such as plasmids and incubation conditions) could be specifically optimized to improve the production capacity of the target protein [37]. Previously, several recombinant fungal immunomodulatory proteins, such as FIP-Lrh from *L. rhinoceros* [22], FIP-gat from *Ganoderma atrum* [38], and FIP-sch2 from *Stachybotrys chlorohalonata* [36] were successfully produced in *E. coli* and later showed efficient bioactivities. In Kong’s study [21], the production of bioactive FIP-fve in E. coli was six times higher than extracted from nature and has competent effects in promoting IFN-γ production in vivo. The successful landing of rFIP-hma heterologous expression provides a practicable approach to enhance its production efficiency and yield. 

### 2.4. Purification of rFIP-hma

The target protein was loaded onto a Ni-NTA column and eluted with different concentrations of imidazole solutions (Figure 5A). In the 250 mM concentration imidazole eluate, the target protein showed a strong expression signal and high purity, meanwhile, the heterogeneous impurity proteins of various molecular weights were basically removed (Lane 8, Figure 5A). SUMO Protease successfully removed the tag (Figure 5B). The molecular weight of the protein without enzyme digestion is 25~35 kDa (Lane 1, Figure 5B), then SUMO Protease successfully separated the Sumo tag (18 kDa) from the FIP-hma protein (14 kDa) (Lane 2, Figure 5B). When 20 mM imidazole was loaded onto the nickel column affinity chromatography, the target protein FIP-hma was distributed in the effluent (Lane 5, Figure 5B). A sodium dodecyl sulfate-polyacrylamide gel electrophoresis (SDS-PAGE) assay was conducted, which confirmed the final molecular weight of the protein was around 14 kDa and the heterobands were eliminated after purification (Lane 5, Figure 5B). The results were consistent with the biological information of FIP-hma containing 145 amino acids. Therefore, 20 mM imidazole eluates were collected to obtain high-purity FIP-hma protein. Combined with the IPTG induction experiments, our results suggest the recombinant protein of FIP-hma (rFIP-hma) was successfully expressed, purified, and identified.

The sample proteins contained minimum LPS, as validated by a commercially available LPS endotoxin ELISA kit. The kit applies the classic double-antibody sandwich pattern with an LPS monoclonal antibody coated on the microplates. The reported LPS concentration in samples was below 1.56 ng/mL, as converted based on the absorption standard curve, which helps exclude the effects of LPS on the subsequent bioactive experiments. 

Based on the known information, the isolation and purification of FIPs from natural mycelium or fruiting bodies are not only time-consuming but also low yield and costly. For example, only 5–10 mg LZ-8 could be isolated from 340 g wet *G. lucidum* fruiting body [10], whereas the production of bioactive proteins could reach several grams per liter in *E. coli* systems [37]. The heterologous expression of the recombinant protein sometimes lacked the mechanism of post-translational modification and was easy to produce inclusion bodies, which required special purification techniques such as affinity chromatography to isolate the target protein [39]. The Ni-NTA nickel-charged affinity resin adopted in this study allows for the isolation of recombinant proteins containing a polyhistidine (6×His) sequence. After purification, the collected elutes were condensed by centrifuge, where the final concentration of rFIP-hma reached 0.5 mg/mL. Our results support that the application of genetic engineering tools and associated protein purification is useful in producing recombinant FIP proteins.

### 2.5. Detecting the Toxicity of rFIP-hma to RAW 264.7 by MTT

The effect of rFIP-hma on the survival rate of RAW 264.7 was detected by the 3-(4,5-dimethylthiazole 2-yl)-2,5-diphenyltetrazolium bromide (MTT) method. After treatment with 12.5–50 μg/mL rFIP-hma, the cell survival rate did not decrease (Figure 6). The experimental results showed that rFIP-hma did not inhibit the proliferation of RAW 264.7, indicating that rFIP-hma has no toxicity to RAW 264.7.

### 2.6. Effect of rFIP-hma on NO Production

NO plays a crucial role in the immune response [40]. The results suggest rFIP-hma at 12.5–50 μg/mL significantly increased NO production in RAW 264.7 cells (Figure 7). The maximum NO production of RAW 264.7 cultured at 50 μg/mL rFIP-hma was 25.37 ± 0.62 μM. The results indicate that rFIP-hma can activate macrophages by promoting NO production.

### 2.7. Effect of rFIP-hma on Cytokine Expression and Release

The isolated rFIP-hma was tested for its immunoregulatory activities. The Western blot results suggest rFIP-hma at 12.5–50 μg/mL induced expression of cytokines with immune activation effects in RAW 264.7 macrophage cells (Figure 8A). IL-6 protein expression was significantly enhanced when treated with 12.5–50 μg/mL or the LPS positive control as compared with PBS placebo, indicating an activation effect on the cytokine central network (Figure 8B). The addition of rFIP-hma also upregulated iNOS expression in a dose-dependent manner (Figure 8C). The iNOS serves as a crucial signal in regulating macrophage activation and differentiation, and its expression was summit when cocultured with 50 μg/mL of FIP-hma among tested doses (Figure 8C). Another interleukin, IL-1β, was increased by rFIP-hma at 12.5 μg/mL and 50 μg/mL (Figure 8D). The Western blot results suggest that there are effects of rFIP-hma on inducing cytokine expression in macrophages. Afterward, we tested cytokine release to further investigate the effects of FIP-hma on immune function. 

The release of three cytokines, IL-6, TNF-α, and IL-1β from RAW 264.7 was measured by ELISA to evaluate the immunomodulatory capacity of rFIP-hma (Figure 8). The results indicate that rFIP-hma significantly increased cytokine release. FIP-hma at 12.5–50 μg/mL induced the secretion of IL-6 and TNF-α in a dose-dependent manner (Figure 9A,B) as compared with the PBS placebo, where the released concentration of TNF-α and IL-6 from RAW 264.7 cells treated with 50 μg/mL of rFIP-hma reached 30,150.7 and 1999.3 pg/mL, respectively (Figure 9A,B). On the other hand, the induced release of IL-1β by rFIP-hma was low, and only 34.27 pg/mL was released when treated with rFIP-hma at 50 μg/mL (Figure 9C). The results suggest that rFIP-hma selectively induced cytokine release from macrophages.

FIPs are fascinating small and heat-stable bioactive proteins that have been extensively explored for their biological activities since their discovery [10,23,41]. Most FIPs exert immunomodulatory and anti-inflammatory effects by inducing IFN-γ, IL-2, IL-4, IL-12, and TNF-α production in peripheral blood mononuclear cells and inhibiting the overproduction of T helper-2 (Th2) cytokines common in an allergy reaction [42]. In this work, we assessed the immunomodulatory effects of the newly discovered FIP-hma on murine macrophage cells. The results supported rFIP-hma was able to enhance the release of IL-6 and TNF-α from RAW 264.7 cells, indicating the activation of the macrophage cells. It has been reported that FIPs derived from different fungi affect various cytokine profiles, where the results are also related to the dosage, molecular modification, or structure applied. For instance, rFIP-Lrh at lower concentrations (4–8 μg/mL) induced significantly higher production of Th1 (IFN-γ, TNF-α) and Th2 (IL-6, IL-4, IL-5, IL-13) cytokines in mice splenocytes, whereas 16 μg/mL rFIP-Lrh induced significantly higher pro-inflammatory cytokines (TNF-α, IL-6, IL-10) [22]. The findings on rFIP-hma favor that it triggered an acute immune response and activation of the central immunoregulatory networking, which were similar to the observed effects of several reported FIPs such as FIP-gsi (isolated from *G. sinensis*) [13] and ACA (isolated from *A. camphorate)* [26]. These results also add evidence that not only the Fve-type FIPs but also other types of FIPs existing in edible and medicinal fungi exhibit immunomodulatory activities. 

As a potent T-cell and macrophage activator, FIPs are considered a good candidate for ingredient development in functional foods and pharmaceuticals. The findings of this study provide an in-depth bioinformatic analysis of FIPs from *H. marmoreus* and an optimized recombinant technique allowing the mass production of rFIP-hma, which showed immune activation activity and might be further applied as an alternative ingredient in biomedicine.

## 3. Materials and Methods

### 3.1. Materials

The *H. marmoreus* strain, Finc-W-247, was collected from Shanghai Finc Bio-tech Inc., Shanghai, China. *H. marmoreus* was sequenced with Pacbio and Illumina [43]. RNAiso Plus, PrimeScript^TM^ RT reagent Kit, and SYBR^®^ Premix Ex Taq II were purchased from TaKaRa (Beijing, China). TOP10 receptor cells were produced by Anhui Global Gene Company (Hefei, China). IPTG was purchased from Amresco (Beijing, China). LPS endotoxin ELISA kit was purchased from Solarbio (Beijing, China). Ni-NTA agarose column was purchased from Smart-Lifesciences (Changzhou, China). Modified Bradford Protein Assay kit was purchased from Sango (Shanghai, China). RAW 264.7 were obtained from the Chinese Academy of Sciences (Shanghai, China). DMEM medium was purchased from Gibco (Shanghai, China). Fetal bovine serum was purchased from Sigma (Port Melbourne, Australia). Penicillin-streptomycin and MTT were purchased from Beyotime (Shanghai, China). LPS was purchased from Sigma (St. Louis, MO, USA). IL-6, TNF-α, and IL-1β ELISA kits were purchased from Solarbio (Beijing, China). Rabbit anti-IL-6 polyclonal antibody, Rabbit anti-iNOS polyclonal antibody, and Rabbit anti-IL-1β polyclonal antibody were purchased from Bioss (Beijing, China).

### 3.2. Bioinformatics Analysis

Amino acid sequences of FIPs, listed in Appendix A, were downloaded from NCBI (https://www.ncbi.nlm.nih.gov (accessed on 9 October 2022)). To identify immunomodulatory protein homologous genes in *H. marmoreus*, c13717 amino acid sequence was used as a bait [44], and a local BLAST search was performed to search for immunomodulatory protein homologous genes from the sequenced *H. marmoreus* genome [43].

Primary structure analyses of FIP-hma were conducted using ProtParam (https://web.expasy.org/protparam/ (accessed on 11 January 2022)) and ProtScale (https://web.expasy.org/protscale/ (accessed on 11 January 2022)) [45] web servers for identifying sequence characteristics. Clustal X2 was used to perform multiple sequence alignment [46]. Invoking the MEGAX64 [47] software package, a neighbor-joining method was used to construct a phylogenetic tree by repeating 500 times of calculation. By using SWISS-MODEL, template sequences with >30% similarity were searched for homology modeling.

### 3.3. Real-Time PCR Assay

Total RNA was extracted by RNAiso Plus. PrimeScript^TM^ RT kit was used to reverse transcribe RNA into cDNA. cDNAs from different developmental stages of *H. marmoreus* were used as templates. We used *α-tubulin* as the housekeeping according to a previous study [48]. A partial actin gene in *H. marmoreus* was cloned using the degenerate PCR primers, α-tubulinF and α-tubulinR. Primers for *FIP-hma* and *α-tubulin* were designed according to their cDNA sequences using Primer Premier 6.0 (Table 1). Then, 2 µL of cDNA template was mixed with 0.8 µL of forward and reverse primers (10 nM), and the reaction system also contained 6 µL ddH_2_O, 0.4 µL ROX, 10 µL TB Green Premix Ex Taq II in the last 20 µL. The loop condition was set as initial degeneration for 30 s at 95 °C; followed by 40 cycles for 5 s at 95 °C; 15 s to 60 °C; and 72 °C for 15 s. Each biological sample was tested in triplicates.

### 3.4. Expression of rFIP-hma

The nucleotide sequence of *FIP-hma* was synthesized by Anhui Huanqiu Gene Technology Co., Ltd., Maanshan, China. Synthesized *FIP-hma* was cloned into the corresponding restriction enzyme sites in pSmartI expression vector, forming the recombinant plasmid pSmartI-FIP-hma.

pSmartI-FIP-hma was transformed into *E. coli* BL21(DE3) cells for the expression of rFIP-hma. Monoclonals were selected and inoculated in Luria-Bertani liquid medium, shaken at 37 °C, 220 rpm until the optical density OD_600_ reached 0.6–0.8. IPTG was added to induce expression. After incubation at 37 °C, 220 rpm for 4 h, the bacterial cells were harvested and disrupted by ultrasonic to release protein. The extracted protein was analyzed by Western blot. The protein and loading buffer were resuspended, boiled for 10 min. After centrifugation, the sample was loaded onto a 12% SDS-PAGE. For further confirmation, the gels were transferred to a PVDF membrane and blocked with 5% skimmed milk powder for 1 h. The mouse anti-His-tag antibody was added at 1:2000 dilution into the skimmed milk powder to react with rFIP-hma. After washing, the membranes were treated with 1:2000 goat anti-mouse antibody. Protein bands were finally developed by TCL chromogenic reagent.

IPTG was added to the medium at concentrations of 0.2 mM and 1.0 mM, respectively, to optimize yield. The bacterial suspension was shaken at 37 °C or 15 °C, 220 rpm to induce fusion protein expression. The soluble proteins were separated by centrifugation at 15,000 rpm, 4 °C for 10 min. All samples were resuspended in SDS-PAGE and stained with Coomassie Brilliant Blue for size (molecular weight) determination.

### 3.5. Purification of rFIP-hma

The proteins were purified by a Ni-NTA agarose column. The bacteria were suspended in Buffer A (20 mM Tris, 300 mM NaCl, 10% Glycerol, pH8.0), and protease inhibitor (PMSF) was added with a final concentration of 1 mM during ultrasonic crushing (Φ 10, 15%, 3 s/6 s, 30 min). After centrifugation at 12,000 rpm for 30 min, the supernatant was collected. A total of 2 mL of Ni NTA Beads were loaded into a gravity column. The samples were added to the balanced Ni NTA Beads, and incubated at 4 °C for 30 min, before collecting the effluent. The column was washed with imidazole solution with the concentrations 20 mM, 40 mM, 60 mM, 80 mM, and 100 mM. The cleaning solutions were collected separately. Finally, the rFIP-hma was eluted with 250 mM imidazole 5–10 times the column volume, and the eluent was collected. After dialysis was completed, a SUMO Protease was mixed with the fusion protein and placed at 4 °C overnight. The mixture was loaded into the Ni-NTA purification column slowly, and the effluent was collected. The purification column was cleaned with Buffer A, and the cleaning solution was collected. The protein was eluted with 20 mM imidazole, and the eluent was collected. Then, the proteins at different concentrations of imidazole were collected by SDS-PAGE analysis. The total rFIP-hma protein mass was detected using the Modified Bradford Protein Assay kit was purchased from Sango (Shanghai, China).

### 3.6. Cell Culture

RAW 264.7 cells were cultured in DMEM medium containing 10% fetal bovine serum and 1% penicillin-streptomycin in a humidified 5% CO_2_ incubator at 37 °C. The 18th passage RAW 264.7 cells were used in experiments.

### 3.7. MTT Assay

The cytotoxic activity of rFIP-hma against RAW 264.7 was determined using the MTT assay. The RAW 264.7 cells with a cell concentration of 6×10^4^ cells/mL were seeded on a 96-well microplate in 5% CO_2_ at 37 °C for 12 h. The medium in the well was discarded. Then, 200 μL of the mixture solution of rFIP-hma (0, 12.5, 25, and 50 μg/mL final concentrations) and culture medium were added. After 24 h of incubation, the culture medium was removed, and 50 μL MTT solution (5 mg/mL) was added to each well for incubation in 5% CO_2_ at 37 °C for 4 h. After 4 h, the MTT solution was removed and the wells were rinsed with PBS. Afterward, 100 μL of DMSO solution was added to each well and the well was placed on a flat plate rotator for 10 min. The absorbance was determined at 490 nm.

### 3.8. NO Determination Assay

The RAW 264.7 cells with a cell concentration of 5 × 10^5^ cells/mL were seeded on a 96-well microplate in 5% CO_2_ at 37 °C for 12 h. The cells were treated with rFIP-hma (12.5, 25, and 50 μg/mL final concentrations) or PBS (10%, as placebo), LPS from *E. coli* O55:B5 (10 μg/mL, as positive control) for 48 h. Then, 100 μL of supernatant was mixed with 50 μL of Griess reagent, and after incubating, the mixed solution was set at room temperature for 10 min. The absorbance was measured at 543 nm.

### 3.9. Effect of rFIP-hma on Cytokine Expression and Release

To evaluate the immunomodulatory bioactivities of rFIP-hma, cytokine expression and release of RAW 264.7 cells were determined by Western blot and ELISA assay. Western blots were performed according to a previously described method [49]. The RAW 264.7 cells with a cell concentration of 3 × 10^5^ cells/mL were seeded on a 6-well microplate in 5% CO_2_ at 37 °C for 8 h. The rFIP-hma protein was dissolved in PBS solution as stocking. The cells were treated with rFIP-hma (12.5, 25, and 50 μg/mL final concentrations) or PBS (10%, as placebo), LPS from *E. coli* O55:B5 (10 μg/mL, as positive control) for 18 h. The proteins were extracted with RIPA buffer containing protease and phosphatase inhibitors. The mixture was centrifugated at 7826 rpm, 4 °C for 15 min. The concentrations of proteins were determined by a BCA protein assay kit. Western blot was conducted as described earlier. The primary antibodies of IL-6, iNOS, IL-1β, and β-actin and specific secondary antibodies were applied. The protein bands were visualized on Tanon 5200 using chemiluminescence (ELC) detection reagent. Signal intensity was analyzed by the Image J-2 software.

The cellular release of IL-6, TNF-α, and IL-1β were determined by ELISA using Mouse IL-6, TNF-α, and IL-1β ELISA Kits. The RAW 264.7 cells with a cell concentration of 3 × 10^5^ cells/mL were seeded on a 96-well microplate in 5% CO_2_ at 37 °C for 8 h. Then, 100 μL of the mixture solution of rFIP-hma (12.5, 25, and 50 μg/mL final concentrations) and culture medium were added. Meanwhile, PBS (10%) and LPS (10 μg/mL) were used as placebo and positive controls, respectively. After 24 h of incubation, the levels of IL-6, TNF-α, and IL-1β in the supernatants were determined using ELISA Kits according to the manufacturer’s instructions, where serial concentrations of murine IL-6 (7.81 to 500 pg/mL) and murine TNF-α, and IL-1β (31.25 to 2000 pg/mL) were used as standards.

### 3.10. Statistical Analysis

All experiments were conducted in triplicates. All experimental data were presented as mean ± standard deviation (SD). The differences between groups were analyzed by ANOVA and Duncan’s multiple-range test using the SPSS22 software. Groups with no letters in common are significantly different (*p* < 0.05). 

## 4. Conclusions

As potent T-cell and macrophage activators, FIPs are considered a good candidate for ingredient development in functional foods and pharmaceuticals. In this study, a new FIP protein was identified in *H. marmoreus* and named FIP-hma. Unlike the most reported Fve-type FIPs, the predicted architecture of FIP-hma showed a conserved CP domain (PF07249) and formed typical spheroid-like structures of Cerato-type FIPs. It showed large evolution divergence from other FIPs according to phylogenetic analysis. rFIP-hma was successfully heterologously expressed in the *E. coli* system and isolated using our designed plasmid and optimized conditions. In murine macrophage cells, rFIP-hma induced expression and extracellular release of central cytokines, such as IL-6 and IL-1β, while no cytotoxic effects were observed according to the MTT assay. Macrophages are principal cells initiating an immune response by the excretion of cytokines. The upregulation of cytokines by rFIP-hma might be associated with downstream immunoregulatory cascades, suggesting an effect of activating the immune response. The FIP-hma has promising potential in the development of dietary supplements, functional foods, and biomedicine. The findings of this study provide an approach to effectively harvest the newly discovered bio-active protein FIP-hma, illustrate its bioinformatics, report its immunoregulatory effects, and shed light on its future applications as an alternative ingredient in biomedicine.

## Figures and Tables

**Figure 1 molecules-28-04796-f001:**
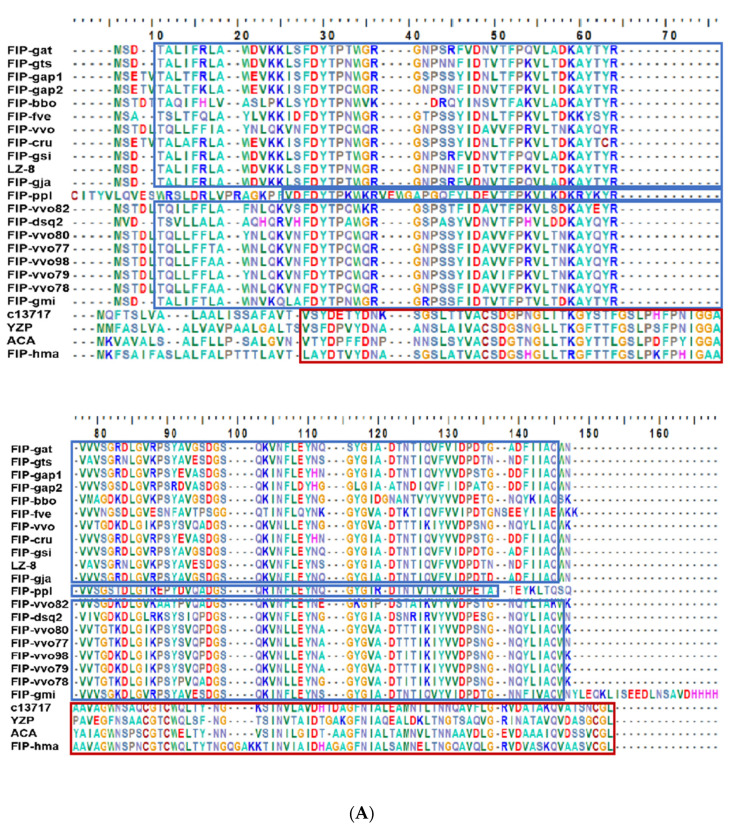
(**A**) Amino acid sequence alignment by Clustal X2. Conserved domain of Fve-type FIPs is included in the blue box. Conserved domain of Cerato-type FIPs is included in the red box. (**B**) Phylogenetic tree of the fungal immunomodulatory proteins.

**Figure 2 molecules-28-04796-f002:**
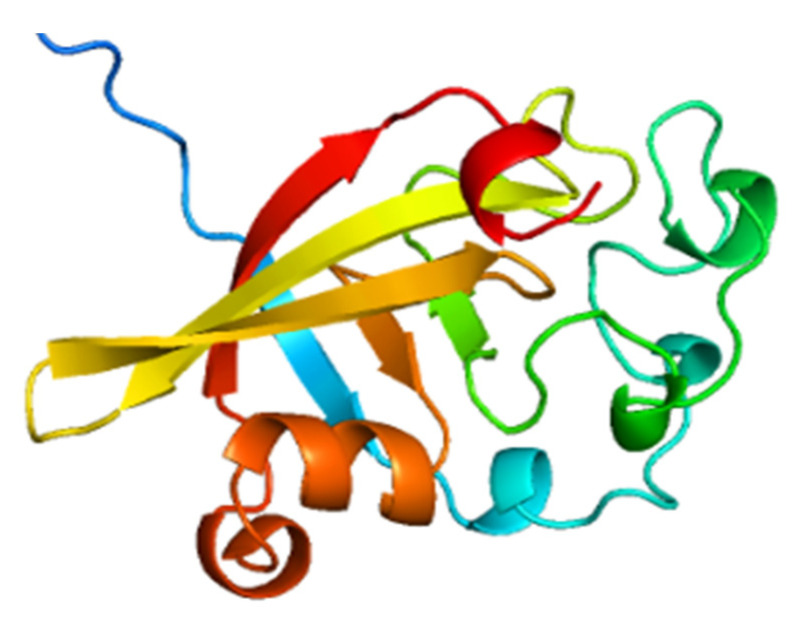
Three-dimensional structure of FIP-hma modeled by AlphaFold.

**Figure 3 molecules-28-04796-f003:**
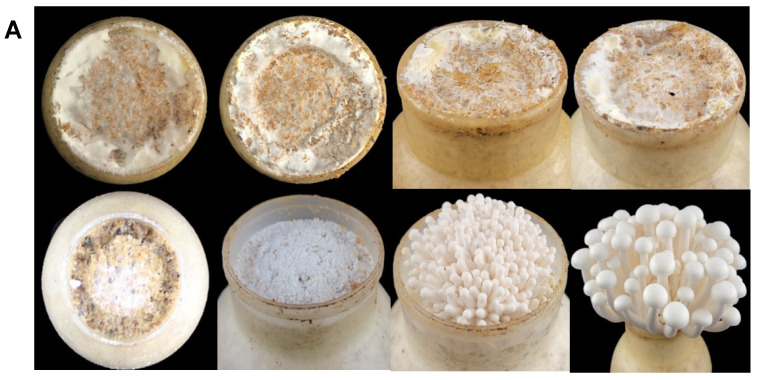
Expression levels of *FIP-hma* in different growth stages and tissues. (**A**) The growth and development of *H. marmoreus*. From left to right and then from top to bottom: 1, half of the mycelia colonized the substrate; 2, mycelia fully colonized the substrate; 3, middle of the post-ripening; 4, completion of the post-ripening; 5, mycelium stimulation; 6, primordia; 7, young fruiting body; 8, mature fruiting body. (**B**) Vegetative growth stages of *FIP-hma*. (**C**) Different tissues in the reproductive growth stages of *FIP-hma*. F, tissue from fruiting body; S, substrate containing mycelium. Multiple comparisons were performed using one-way analysis of variance (ANOVA) combined with Duncan’s test to assess significant differences between mean values of groups (*p* < 0.05). Groups sharing no common letter are significantly different.

**Figure 4 molecules-28-04796-f004:**
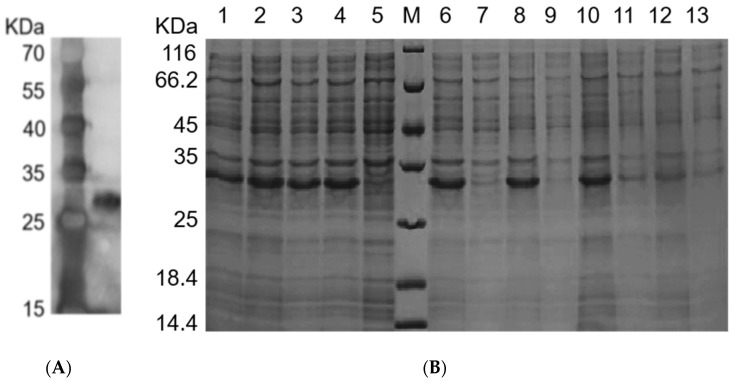
Expression and optimization of rFIP-hma. (**A**) Western blot analysis of expression rFIP-hma. Lane M, protein marker; Lane 1, sample treated with IPTG. (**B**) SDS-PAGE analysis of rFIP-hma. Lane M, protein marker; Lane 1, sample after treatment with IPTG at 15 °C 0.2 mM; Lane 2, sample after treatment with IPTG at 15 °C 1.0 mM; Lane 3, sample after treatment with IPTG at 37 °C 0.2 mM; Lane 4, sample after treatment with IPTG at 37 °C 1.0 mM; Lane 5, sample without IPTG treatment; Lane 6, precipitated sample after treatment with IPTG at 37 °C 1.0 mM; Lane 7, supernatant sample after treatment with IPTG at 37 °C 1.0 mM; Lane 8, precipitated sample after treatment with IPTG at 37 °C 0.2 mM; Lane 9, supernatant sample after treatment with IPTG at 37 °C 0.2 mM; Lane 10, precipitated sample after treatment with IPTG at 15 °C 1.0 mM; Lane 11, supernatant sample after treatment with IPTG at 15 °C 1.0 mM; Lane 12, precipitated sample after treatment with IPTG at 15 °C 0.2 mM; Lane 13, supernatant sample after treatment with IPTG at 15 °C 0.2 mM.

**Figure 5 molecules-28-04796-f005:**
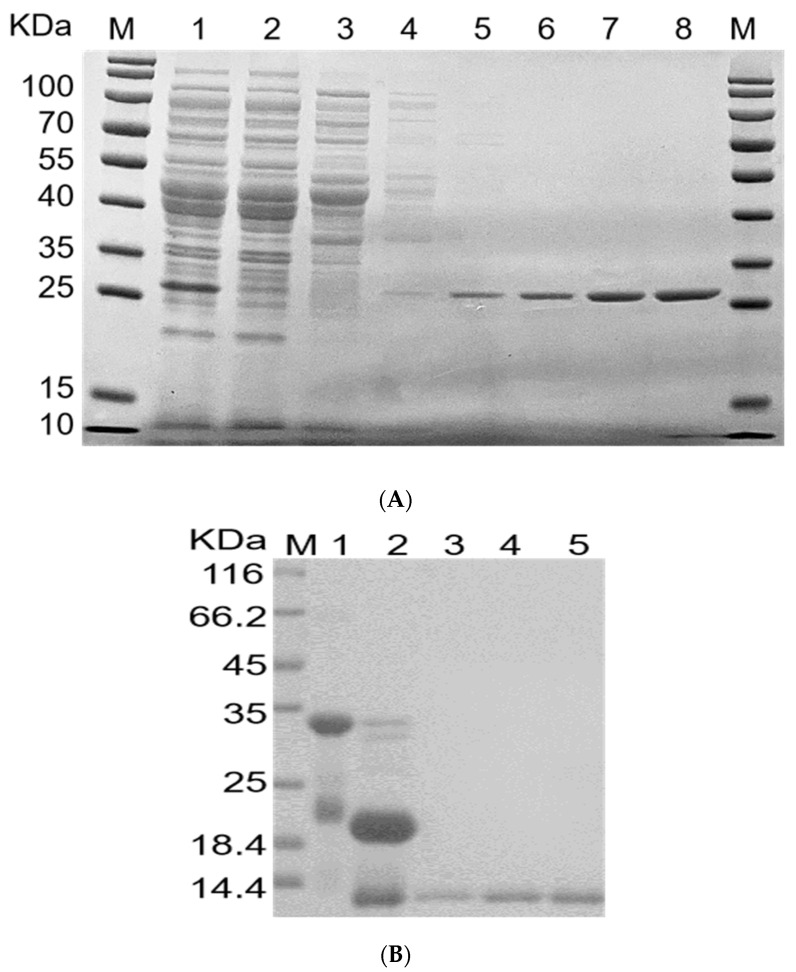
Purification of rFIP-hma. (**A**) SDS-PAGE of purified rFIP-hma. Lane M, protein marker; Lane 1, supernatant after crushing; Lane 2, effluent sample; Lane 3, 20 mM imidazole eluent; Lane 4, 40 mM imidazole eluent; Lane 5, 60 mM imidazole eluent; Lane 6, 80 mM imidazole eluent; Lane 7, 100 mM imidazole eluent; Lane 8, 250 mM imidazole eluent. (**B**) SDS-PAGE analysis of rFIP-hma enzyme digestion. Lane M, protein marker; Lane 1, sample before enzyme digestion; Lane 2, digested sample after enzyme digestion; Lane 3, efluate sample; Lane 4, Buffer A effluent sample; Lane 5, 20 mM imidazole eluent.

**Figure 6 molecules-28-04796-f006:**
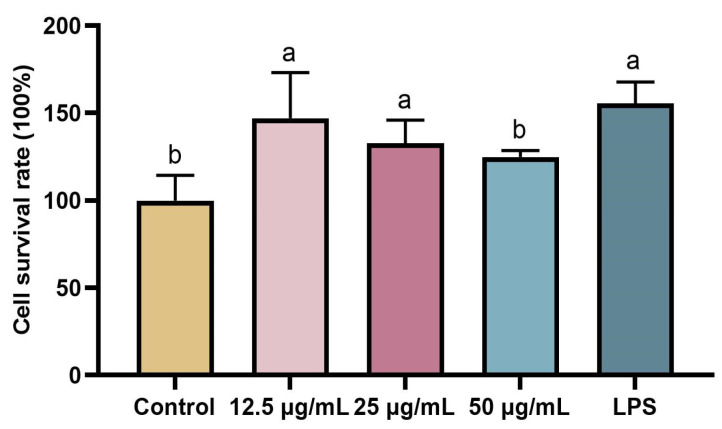
The effect of rFIP-hma on the survival rate of RAW 264.7 by MTT. Multiple comparisons were performed using one-way analysis of variance (ANOVA) combined with Duncan’s test to assess significant differences between mean values of groups (*p* < 0.05). Groups sharing no common letter are significantly different.

**Figure 7 molecules-28-04796-f007:**
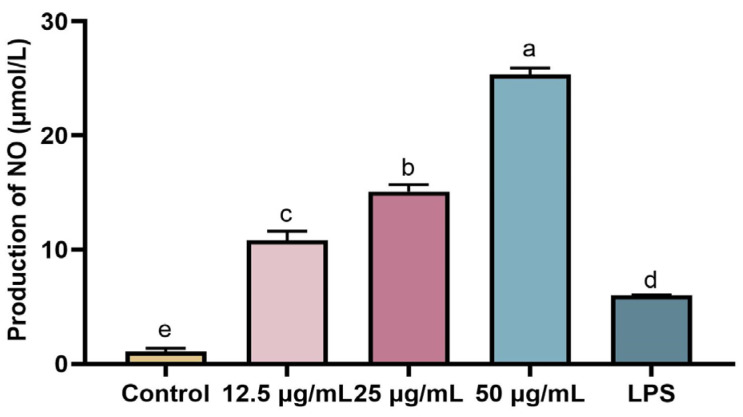
Effect of rFIP-hma on NO production. Multiple comparisons were performed using one-way analysis of variance (ANOVA) combined with Duncan’s test to assess significant differences between mean values of groups (*p* < 0.05). Groups sharing no common letter are significantly different.

**Figure 8 molecules-28-04796-f008:**
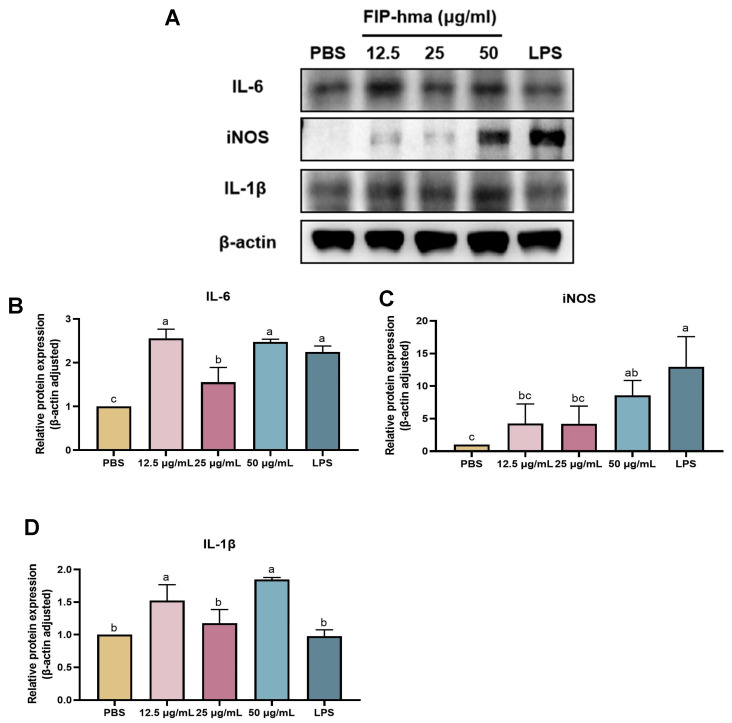
Effects of rFIP-hma on cytokine expression from RAW 264.7 by Western blot. (**A**) Cytokine protein expression was detected by Western blot (**B**) IL-6. (**C**) iNOS. (**D**) IL-1β. PBS was used as the placebo control; LPS (10 μg/mL) was used as the positive control. Multiple comparisons were performed using one-way analysis of variance (ANOVA) combined with Duncan’s test to assess significant differences between mean values of groups (*p* < 0.05). Groups sharing no common letter are significantly different.

**Figure 9 molecules-28-04796-f009:**
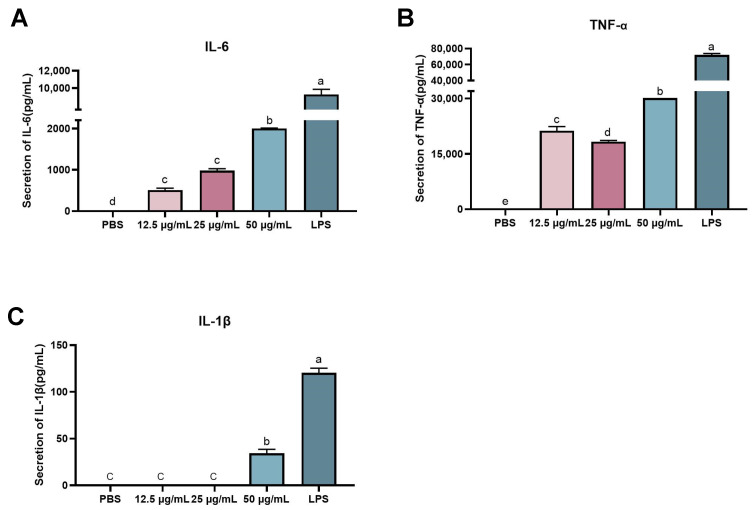
Effects of rFIP-hma on cytokine release from RAW 264.7 by ELISA. (**A**) IL-6 assay. (**B**) TNF-α assay. (**C**) IL-1β assay. PBS was used as the placebo control; LPS (10 μg/mL) was used as the positive control. Multiple comparisons were performed using one-way analysis of variance (ANOVA) combined with Duncan’s test to assess significant differences between mean values of groups (*p* < 0.05). Groups sharing no common letter are significantly different.

**Table 1 molecules-28-04796-t001:** Primers used in this study.

Number	Primer Name	Primer Sequence (5′-3′)
1	α-tubulinF	TTGTCCAACACCACCGCCATC
2	α-tubulinR	TGCCGACCTCCTCATAGTCCTT
3	FIP-hmaF	GGACATGCTGGCAGCTCACTTA
4	FIP-hmaR	TCAGAGCCCGCACACAGATG

## Data Availability

Not applicable.

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
