# Peer review of "Characterization, Recombinant Production, and Bioactivity of a Novel Immunomodulatory Protein from Hypsizygus marmoreus"

_molecules, 2023, doi:10.3390/molecules28124796_

Round 1

Reviewer 1 Report

The manuscript explores the properties and the functional effects of FIP-hma, a protein isolated from the edible mushroom Hypsizygus marmoreus. The idea is interesting and of high interest, however, several parts of the manuscript needs to be addressed before considering its publication. A main critical point needs to be addressed by the authors as mentioned below

Critical point:

The authors produced the protein of interest in E. coli and then used it for cytokines production by macrophages. Proteins expressed in bacteria are always contaminated with LPS. The latter is an activator of macrophages and the authors even used it as a positive control. The cytokines production experiment needs to be repeated using LPS-free proteins.

Major points:

Please reformulate the title. The word protein is mentioned twice. Also the words Immunoregulatory and Immunomodulatory refer to the same concept. The title should be concise and points directly to the main subject and findings of the paper.

The conclusion doesn’t valorise the output of this work. The authors needs to reformulate it with emphasis on the immunomodulatory effects of the protein and its potential application and not how they express and purify it.

Line 81: How did the authors conclude that the protein is stable. To what did they compare the instability index? Clarification and references are needed.

The parts 2.1 and 2.2 of results and discussion can be combined into one paragraph entitled: Structural and biochemical properties of FIP-hma. Also, Table 1 can be in supplementary materials. Anyone can get this information using Protparam

The title of figure 2 should be more elaborated by mentioning how the structure was modelled and also divided in A and B.

Line 159 to line 160: ‘liquid fermentation might be a more effective strategy for their protein production’. The authors should explain and develop this observation using references.

Figure 3 should be modified following two axes. The first one is related to the statistical analysis. Indications using stars (*) should be used. For example, * for p<0.05, ** for p<0.01, *** for p<0.001 and ns for non-significant differences. The second axe is the illustration of the expression dynamics in different growth stages. The authors should show illustration of the different studied growth stage or tissue in addition to the shown graph.

Figure 5 B. What’s the two bands observed in lane 1? And in lane 2, what’s the massive band at around 19 kDa?

Line 374: The nucleotide sequence and not the amino acid sequence.

Other minor points to be addressed:

Line 27: develop and not develop

Line 28: reported FROM mushrooms

Line 29: Small molecule compounds is a wrong term. Better use low molecular weight compounds/molecules

Line 48: was found

Line 54: Recombinantly expressed instead of has been recombined and expressed

Line 55: to natural what? Please refer to the natural protein

Line 55 to line 56: FIP-fve was successfully expressed in E. coli and induced to release IL-2 and IFN- γ. Please reformulate

Line 63 to line 64: The sentence needs to be reformulated

Author Response

Dear editor and reviewer

Thank you very much for the suggestions. Please see the attachment. We have considered all the comments made by you and referees and have modified our paper in response to each comment. We have listed each comment and our response on an attached sheet. We believe that these changes have significantly improved the paper.

Best regards

Reviewer 2 Report

The authors present an interesting study of a new immunomodulatory protein from the fungus Hypsizygus marmoreus. As demonstrated for other fungal immunomodulatory proteins (FIP), the properties of FIP-hma could be of medicinal interest.

In this manuscript, the authors identified a FIP in H. marmoreus, which they called FIP-hma. Using bioinformatics, they predicted the physiological and biochemical characteristics of FIP-hma. Also, they determined the expression levels of FIP-hmas during the growth and development of H. marmoreus. To succeed in the aim of this study, the expression and purification of FIP-hma were performed to test the immune regulation activity of the purified rFIP-hma.

After reading the manuscript, important issues must be considered for publishing in Molecules.  

1.     Currently, AlphaFold is the most accepted software to predict 3D structures from the sequence of amino acids. Authors should use it to predict the FIP-hma 3D structure instead of SWISS-MODEL.

2.     Although the authors claim that rFIP-hma is soluble, the results show it mainly in inclusion bodies. Authors have to demonstrate how the rFIP-hma was solubilized from the inclusion bodies.

3.     The most important is demonstrating that LPS from E. coli is absent in the recombinant FIP-hma. If not, the findings of immunomodulatory effects cannot be attributable to FIP-hma.

Author Response

(The authors gave the same response as above.)

Reviewer 3 Report

In the paper entitled: Characterization, Recombinant Protein Production and Immunoregulatory Activity of a Novel Immunomodulatory Protein from Hypsizygus marmoreus, the authors discovered in the genome of the edible mushroom Hypsizygus marmoreus a novel fungal immunomodulatory protein (FIP), named as FIP-hma. Bioinformatics analysis indicated that FIP-hma contained the cerato-platanin (CP) conserved domain and was categorized into Cerato- type FIP. Additionally, the authors cloned the cDNA sequence of FIP-hma and successfully expressed this protein in Escherichia coli (E. coli) BL21(DE3), purified, and isolated by Ni-NTA and SUMO-Protease. Recombinant FIP-hma showed immunoregulatory activities. In the RAW 264.7 macrophages, the expression levels of TNF- 22 α, IL-1β, and IL-6 were significantly up-regulated and the release of IL-6 and IL-1β into the extracellular fluid was significantly enhanced indicating its effect of activating an immune response.

      The title of the paper is brief and informative. In the Introduction, the importance of the topic discussed in this paper was marked, especially in the context of the role of bio-active ingredients present in edible mushrooms, and also fungal immunomodulatory proteins. However, I suggest adding a separate section describing the role of macrophages in the regulation of the immune response and their key functions.

In my opinion, the manuscript can be reconsidered for publication after major revision.

 Main comments:

1. Abstract. Please, correct the sentence (lines 22-25):  In the RAW 264.7 macrophages, the expression levels of TNF- 22 α, IL-1β, and IL-6 were significantly improved and the release of IL-6, iNOS, IL-1β into the extracellular fluid were significantly enhanced by rFIP-hma, indicating its effect of activating immune response’.  iNOS is an intracellular enzyme (that produces NO) and is not secreted into the extracellular fluid.

2. #line 346#  Mention the ‘g’ against ‘rpm’. Check for further in the manuscript throughout.

3. Point  3.5. Purification of rFIP-hma should be described more clearly. Please correct it. Did you check the antibacterial purity (LPS contamination) of recombinant protein rFIP-hma before RAW  264.7 cells stimulation? It is extremely important when you use o E. coli as the expression system.

 4. Point 3.6. Cell Culture. Please add the information about passages of RAW 264.7 that were used in experiments.

 5. Point 3.7. Effect of rFIP-ham on cytokine expression and release.  Please add the RAW cell concentration used in the test. What kind of LPS was used as a positive control (source, strain)? Please explain, why a high dose of 10 ug/ml was used to stimulate RAW macrophages. It is a very high concentration. Did you test the toxicity of rFIP-ham using the MTT test (or other)? It is crucial to be sure, that your protein is not toxic to cells. This needs to be done and added to the paper. The impact of rFIP-ham on iNOS expression was also determined by WB. Did you check if it corresponds to nitric oxide production?

6. Please mention the statistical test used in Figure Legends 3 and 6.

 7. Did you ever compare the biological activity (cytokine expression/production) of FIP-hma isolated from Hypsizygus marmoreus directly, with recombinant rFIP-hma? In my opinion, it is crucial to test the activity of isolated and purified FIP-hma previously, and later recombinant ones.

8. English grammar should be checked once again and corrected.

Author Response

(The authors gave the same response as above.)

Round 2

Reviewer 1 Report

 Manuscript can be accepted after a thorough editing of English language 

Reviewer 2 Report

The authors adequately addressed all comments and requests for corrections.

Reviewer 3 Report

 The manuscript has been revised according to my comments.